# On the performance improvement of Butterfly Optimization approaches for global optimization and Feature Selection

**Adel Saad Assiri** *

Department of Management Information Systems, College of Business, King Khalid University, Abha, Saudi Arabia

* adaseri@kku.edu.sa

**Data Availability Statement:** BOA: https://www. mathworks.com/matlabcentral/fileexchange/ 68209-butterfly-optimization-algorithm-boa?s_tid= prof_contriblnk PSO: https://www.mathworks. com/matlabcentral/fileexchange/67429-a-simple-

## Abstract

Butterfly Optimization Algorithm (BOA) is a recent metaheuristics algorithm that mimics the behavior of butterflies in mating and foraging. In this paper, three improved versions of BOA have been developed to prevent the original algorithm from getting trapped in local optima and have a good balance between exploration and exploitation abilities. In the first version, Opposition-Based Strategy has been embedded in BOA while in the second Chaotic Local Search has been embedded. Both strategies: Opposition-based & Chaotic Local Search have been integrated to get the most optimal/near-optimal results. The proposed versions are compared against original Butterfly Optimization Algorithm (BOA), Grey Wolf Optimizer (GWO), Moth-flame Optimization (MFO), Particle warm Optimization (PSO), Sine Cosine Algorithm (SCA), and Whale Optimization Algorithm (WOA) using CEC 2014 benchmark functions and 4 different real-world engineering problems namely: welded beam engineering design, tension/compression spring, pressure vessel design, and Speed reducer design problem. Furthermore, the proposed approches have been applied to feature selection problem using 5 UCI datasets. The results show the superiority of the third version (CLSOB-BOA) in achieving the best results in terms of speed and accuracy.

## Introduction

In recent years, the complexity of real-world engineering optimization problems has been increased rapidly due to the advent of the latest technologies. In order to find the optimal solutions to these problems, many optimization methods have been introduced to find the optimal solutions. These algorithms can be divided into 2 major categories: deterministic and stochastic. In the formal category, for example Linear and non-linear programming [1], the solution of the current iteration is used in the next iteration to get the updated solution. The methods in this category have some limitations such as falling into local optima, single based solutions, and other issues regarding search space as mentioned in [2]. In the latter category stochastic methods, also known as metaheuristics, which generate & use random variables. This category has many advantages such as flexibility, simplicity, gradient-free and independently to the

implementation-of-particle-swarm-optimization-pso-algorithm?s_tid=prof_contriblnk SCA: https://www.mathworks.com/matlabcentral/fileexchange/54948-sca-a-sine-cosine-algorithm?s_tid=prof_contriblnk MFO: https://www.mathworks.com/matlabcentral/fileexchange/52269-moth-flame-optimization-mfo-algorithm?s_tid=prof_contriblnk WOA: https://www.mathworks.com/matlabcentral/fileexchange/55667-the-whale-optimization-algorithm?s_tid=prof_contriblnk.

**Funding:** King Khalid University.

**Competing interests:** The authors have declared that no competing interests exist.

problems. Metaheuristics algorithms have been proposed by studying creatures' behavior, physical phenomena, or evolutionary concepts and has been successfully applied to many applications [3–5]. Genetic Algorithm (GA) [6], Differential Evolution (DE) [7], Particle Swarm Optimization (PSO) [8], Artificial Bee Colony (ABC) [9], Ant Colony Algorithm (ACO) [10], and Simulated Annealing (SA) [11] are some of the most conventional metaheuristics algorithms. Recently, numerous number of optimization algorithms have been appeared such as: Cuckoo Search (CS) [12], Gravitational Search Algorithm (GSA) [13], Crow Search Algorithm (CSA) [14], Dragonfly Algorithm (DA) [15], Biogeography-Based Optimization algorithm (BBO) [16], Bat algorithm (BA) [17], Whale Optimization Algorithm (WOA) [18], Grasshopper optimization algorithm (GOA) [19], Emperor penguin optimizer (EPO) [20], Squirrel search algorithm (SSA) [21], Seagull Optimization Algorithm (SOA) [22], Nuclear Reaction Optimization (NRO) [23], Salp swarm algorithm [24], Harris Hawks Optimization (HHO) [25], Slime Mould Algorithm (SMA) [26], Henry Gas Solubility Optimization (HGSO) [27], Elephant Herd Optimization (EHO) [28], Ant-Lion Optimization (ALO) [29] and Moth-Flame Optimization (MFO) [30].

Butterfly optimization algorithm [31] is a novel population-based metaheuristics algorithm that mimics butterflies foraging behavior. BOA has been applied to many fields. In [32] Aygül et al. use BOA to find the maximum power point tracking under partial shading condition (PSC) in photovoltaic (PV) systems. Lal et al. in [33] presented Automatic Generation Control (AGC) to 2 nonlinear power systems using BOA. Also, in [34] Arora and Anand embedded learning automata in BOA. Li et al. in [35] proposed an improved version of BOA using Cross-Entropy method to achieve a better balance between exploration and exploitation. Arora and Anand proposed a binary version of BOA and applied it to the Feature Selection (FS) problem [36]. Another binary version which also applied to feature selection is introduced by Zhang et al. [37] by using new initialization strategy and new operator has been added to transfer function. Likewise, Fan et al. [38] tried to improve BOA performance by adding fragrance coefficient and enhancing local & global search.

A guiding weight and population restart are done by Guo et al. [39]. BOA has been also hybridized with other metaheuristics algorithms such as FPA [40] and ABC [41]. Also, Sharama and Saha in [42] proposed an updated version of BOA using mutualism scheme. In spite of, many real-world problems have been solved by using the original BOA due to its advantages as easy in implementation, simplicity, less number of parameters. However, in some cases like other MH algorithms, it may stuck in local optima regions which lead to premature convergence problems.

However, the success of the above mentioned algorithms in enhancing BOA search capabilities, it still have some limitations and drawbacks: 1) BOA still have difficulties to escape from local optimum region especially when BOA is applied to complex or high dimensional problems. 2) all enhanced BOA variants solve only one problem (Initialization, diversity, and balancing between exploration & exploitation). This encourages and motivates us to introduce some other enhancement.

Opposition-based Learning strategy (OBL) has been integrated with many MH algorithms like PSO [43], GSA [44], ACO [45], GWO [46] and DE [47] to strength their exploration abilities. Also, Chaotic Local Search (CLS) strategy is used in order to make a good balance between exploration and exploitation. CLS concepts was introduced in numerous number of MHs such as PSO [43], Tabu search [48] and ABC [49].

In this paper, three enhanced versions of BOA has been introduced. In the first proposed version Opposition-based Learning strategy is used to enhance the population diversity by checking the opposite random solutions in the initialization phase and the updating step. In the second proposed version, Chaotic Local Search (CLS) has been incorporated in BOA to

exploits the regions near to the best solutions. In the last version, both of OBL and CLS are used together to enhance overall performance. To best of our knowledge, this is the first time to use CLS, OBL concepts in BOA algorithms.

This paper is organized as follows: section 2 provides the basics of BOA. The three novel variants and the concepts of OBL & CLS are introduced in section 3. the experiments results & Discussion and Conclusion & Future work are shown in sections 4 and 5 respectively.

## 1 Preliminaries

In this section, the BOA inspiration and mathematical equations are shown first. Then, the basics of Opposition-based Learing and Chaotic Local Search are presented.

### 1.1 Butterfly optimization algorithm

The BOA equations and complexity is described in details in the following subsections.

**1.1.1 Inspiration & mathematical equations.** Butterflies belong to the Lepidoptera class in the Animal Kingdom Linnaean system [50]. In order to find food/mating partner, they used their sense, sight, taste, and smell. Butterfly Optimization Algorithm (BOA) is a recent nature-based algorithm developed by Arora and Singh in 2018 [31]. BOA simulates the behavior of butterflies in food foraging. Biologically, each butterfly has sense receptors that cover all butterfly's body. These receptors are considered chemoreceptors and are used in smelling/sensing the food/flower fragrance. To model butterflies' behavior, it's assumed that each butterfly produce fragrance with some power/intensity. if a butterfly is able to sense fragrance from the best butterfly, it moves towards the position of the best butterfly. On the other hand, if a butterfly can't sense fragrance, it moves randomly in the search space. In BOA, the fragrance is defined as a function of physical intensity as given in 1.

$$pf_i = cI^a \tag{1}$$

where $pf_i$ refers to the amount of fragrance perceived by other butterflies, c is the sensory modality, I and a refer to stimulus intensity and power exponent respectively. Global search (exploration) and local search (exploitation) phases are given respectively by Eqs 2 and 3.

$$x_i(t+1) = x_i(t) + (r^2 \times g^* - x_i(t)) \times pf_i \tag{2}$$

$$x_i(t+1) = x_i(t) + (r^2 \times x_j(t) - x_k(t)) \times pf_i \tag{3}$$

**Algorithm 1** Butterfly Optimization Algorithm (BOA)
```
 1: Initialize Dim, Max_Iter, curr_Iter, Objective Function
 2: Generate a uniform distributed solutions (Initial Population)
      X = (x₁, x₂, ..., xₙ)
 3: Define sensory modality c, stimulus intensity I, and switch
      probability p
 4: calculate stimulus intensity Iᵢ at xᵢ using f(xᵢ)
 5: while (curr_Iter ¡ Max_Iter) do
 6:   for each butterfly in (X) do
 7:     Calculate fragrance using Eq 1
 8:   end for
 9:   g* = best butterfly
10:   for each butterfly in (X) do
11:     r = rand()
12:     if r ¡ p then
13:       Update butterfly position using Eq 2
14:     else
```

```
15:      Update butterfly position using Eq 3
16:    end if
17:  end for
18:  Update value of a
19: end while
20: Return g*.
```

**1.1.2 Complexity.** To be able to compute the BOA complexity, assume the population size is ($P$), maximum iteration number ($N$), the problem dimensions ($D$). Then, the BOA complexity can be calculated as follows $O(N(D \times P + D \times C))$ where $C$ refers to the cost of the fitness function $= O(NDP + NDC)$.

## 1.2 Opposition-based Learning

Tizhoosh in [51] introduced Opposition-based learning (OBL) to accelerate the convergence by calculating the opposite solution of the current one and taking the best of them. In [47] a mathematical proof is given to show that the opposite solutions are more likely to be near optimal than totally random. The opposite solution $\bar{X}_i$ can be calculated from the following equation

$$\bar{X}_i = a + b - X_i, X_i \in [a, b] \tag{4}$$

where a, b is the lower bound and the upper bound respectively.

## 1.3 Chaotic local search

Chaotic system characteristic can be used to make local search operator in order to strengthen the exploitation abilities in solving optimization tasks. Chaos is based on the navigation of deterministic nonlinear complex behavior. There are many chaotic maps in literature such as logistic, singer, tent, piecewise, and sinusoidal. This is because of the efficiency of chaotic map is related to the problem itself as mentioned by Fister et al. [52, 53]. Logistics map is used in this paper and its sequequence can be obtained from the following equation.

$$C_{i+1} = \mu \times C_i \times (1 - C_i), i = 1, 2, ..., n - 1 \tag{5}$$

where $\mu = 4$, set $0 \leq C_1 \leq 1$ and $C_1 \neq 0.25, 0.5, 0.75, 1$. To calculate the candidate solution CS from the target position T, the next equation is used.

$$CS = (1 - s) \times T + S \times \acute{C}_i, i = 1, 2, ..., n - 1 \tag{6}$$

## 2 The proposed approaches

### 2.1 Opposition-Based BOA (OBBOA)

The first version is called OBBOA which improves the performance of BOA by using OBL strategy. OBL enhance the BOA algorithm by improving its ability to explore search space deeply and speed up the reaching to optimal value. This version consists of 2 stages: First, at the initialization stage by calculating the opposite solution to each one in the initialization, then selecting the best N values. Second OBL is embedded in the updating stage. The pseudo-code of this version is given in Alg. 2.

**Algorithm 2** Opposition-Based BOA (OBBOA)

```
1: Initialize Dim, Max_Iter, curr_Iter, Objective Function
2: Generate a uniform distributed solutions (Initial Population)
      X = (x₁, x₂, ..., xₙ)
```

```
 3: Define sensory modality c, stimulus intensity I, and switch
    probability p
 4: calculate stimulus intensity I_i at x_i using f(x_i)
 5: Compute X̄
 6: Select best N from X ∪ X̄
 7: while (curr_Iter < Max_Iter) do
 8:   for each butterfly in (X) do
 9:     Calculate fragrance using Eq 1
10:   end for
11:   g* = best butterfly
12:   for each butterfly in (X) do
13:     r = rand()
14:     if r ≤ p then
15:       Update butterfly position using Eq 2
16:     else
17:       Update butterfly position using Eq 3
18:     end if
19:     Calculate x̄
20:     x_i = x̄_i if f(x_i)) < f(x̄_i)
21:   end for
22:   Update value of a
23: end while
24: Return g*.
```

## 2.2 Chaotic Local Search BOA (CLSBOA)

In the second version which is called CLSBOA, Chaotic Local Search is integrated with BOA to make a proper balance between exploration and exploitation. The pseudo-code of this version is introduced in Alg. 3.

**Algorithm 3** Chaotic Local Search BOA (CLSBOA)

```
 1: Initialize Dim, Max_Iter, curr_Iter, Objective Function
 2: Generate a uniform distributed solutions (Initial Population)
      X = (x_1, x_2, ..., x_n)
 3: Define sensory modality c, stimulus intensity I, and switch
    probability p
 4: calculate stimulus intensity I_i at x_i using f(x_i)
 5: while (curr_Iter < Max_Iter) do
 6:   for each butterfly in (X) do
 7:     Calculate fragrance using Eq 1
 8:   end for
 9:   g* = best butterfly
10:   for each butterfly in (X) do
11:     r = rand()
12:     if r < p then
13:       Update butterfly position using Eq 2
14:     else
15:       Update butterfly position using Eq 3
16:     end if
17:   end for
18:   Generate the candiate solution CS by performing CLS strategy
19:   g* = CS if f(CS) < f(g*)
20:   Update value of a
21: end while
22: Return g*.
```

### 2.3 Chaotic Local Search Opposition-Based BOA (CLSOBBOA)

In this version, both of the 2 previous modification has been added together to enhance BOA and get the most near optimal solution.

Complexity:

To be able to compute the BOA complexity, assume the population size is ($P$), maximum iteration number ($N$), the problem dimensions ($D$). Then, the CLSOBBOA complexity can be calculated as follows $O(BOA) + O(OBL) + O(CLS) = O(N(D \times P + D \times C + P + P))$ where $C$ refers to the cost of the fitness function = $O(NDP + NDC)$

**Algorithm 4** Chaotic Local Search & Opposition-Based BOA (CLSOBBOA)

```
 1: Initialize Dim, Max_Iter, curr_Iter, Objective Function
 2: Generate a uniform distributed solutions (Initial Population)
       X = (x₁, x₂, ..., xₙ)
 3: Define sensory modality c, stimulus intensity I, and switch
      probability p
 4: calculate stimulus intensity Iᵢ at xᵢ using f(xᵢ)
 5: Compute X̄
 6: Select best N from X ∪ X̄
 7: while (curr_Iter ¡ Max_Iter) do
 8:   for each butterfly in (X) do
 9:      Calculate fragrance using Eq 1
10:   end for
11:   g* = best butterfly
12:   for each butterfly in (X) do
13:      r = rand()
14:      if r ¡ p then
15:         Update butterfly position using Eq 2
16:      else
17:         Update butterfly position using Eq 3
18:      end if
19:      Calculate x̄
20:      xᵢ = x̄ᵢ if f(x₍ᵢ₎) < f(x̄ᵢ)
21:   end for
22:   Generate the candiate solution CS by performing CLS strategy
23:   g* = CS if f(CS)<f(g*)
24:   Update value of a
25: end while
26: Return g*.
```

## 3 Experiments

In this section, the proposed algorithms are tested using CEC as shown in the first subsection after that these algorithms are applied to 5 UCI datasets.

### 3.1 Benchmark functions

In this subsection, 30 functions have been used to compare algorithms using many statistical measure.

**3.1.1 Test functions.** A set of 30 functions from CEC 2014 are used to compare the performance of the proposed algorithms with other state-of-art algorithms. This benchmark functions have new characteristics such as rotated trap problems, graded level of linkage, and composing functions through dimensions-wise properties. This benchmark can be categorized to the following (Unimodal, Multi-modal, Hybrid, and Composite functions) and the definition of these function can be shown in Table 1 where opt. refers to the mathematical optimal value and the bound of the variables in the search space falls in the interval $\in [-100, 100]$.

**Table 1. CEC2014 functions.**

| No. | Types | Name | Opt. |
|---|---|---|---|
| F1(CEC) | Unimodal fnctions | Rotated high conditioned elliptic function | 100 |
| F2(CEC) | | Rotated bent cigar function | 200 |
| F3(CEC) | | Rotated discus function | 300 |
| F4(CEC) | Simple multimodal functions | Shifted and rotated Rosenbrocks function | 400 |
| F5(CEC) | | Shifted and rotated Ackleys function | 500 |
| F6(CEC) | | Shifted and rotated Weierstrass function | 600 |
| F7(CEC) | | Shifted and rotated Griewanks function | 700 |
| F8(CEC) | | Shifted Rastrigins function | 800 |
| F9(CEC) | | Six Hump Camel Back | 900 |
| F10 (CEC) | | Shifted and rotated Rastrigins function | 1000 |
| F11 (CEC) | | Shifted and rotated Schwefels function | 1100 |
| F12 (CEC) | | Shifted and rotated Katsuura function | 1200 |
| F13 (CEC) | | Shifted and rotated HappyCat function | 1300 |
| F14 (CEC) | | Shifted and rotated HGBat function | 1400 |
| F15 (CEC) | | Shifted and rotated Expanded Griewanks plus Rosenbrocks function | 1500 |
| F16 (CEC) | | Shifted and rotated Expanded Scaffers F6 function | 1600 |
| F17 (CEC) | Hybrid functions | Hybrid function 1 (N = 3) | 1700 |
| F18 (CEC) | | Hybrid function 2 (N = 3) | 1800 |
| F19 (CEC) | | Hybrid function 3 (N = 4) | 1900 |
| F20 (CEC) | | Hybrid function 4 (N = 4) | 2000 |
| F21 (CEC) | | Hybrid funcion 5 (N = 5) | 2100 |
| F22 (CEC) | | Hybrid function 6 (N = 5) | 2200 |
| F23 (CEC) | Composition functions | Composition function 1 (N = 5) | 2300 |
| F24 (CEC) | | Composition function 2 (N = 3) | 2400 |
| F25 (CEC) | | Composition function 3 (N = 3) | 2500 |
| F26 (CEC) | | Composition function 4 (N = 5) | 2600 |
| F27 (CEC) | | Composition function 5 (N = 5) | 2700 |
| F28 (CEC) | | Composition function 6 (N = 5) | 2800 |
| F29 (CEC) | | Composition function 7 (N = 3) | 2900 |
| F30 (CEC) | | Composition function 8 (N = 3) | 3000 |

**Table 2. Meta-heuristic algorithms parameters settings.**

| Alg. | Parameter | Value |
|------|-----------|-------|
| BOA | a | 0.1 |
| GWO | a | [0, 2] |
| MFO | t | [−1, 1] |
|  | b | 1 |
| PSO | wMaxt | 0.9 |
|  | wMin | 0.2 |
|  | $c_1$ | 2.0 |
|  | $c_2$ | 2.0 |
| SCA | a | 2 |
| WOA | a | 2 |
|  | b | 2 |

**3.1.2 Comparative algorithm.** In order to test our algorithms, we compare the 3 proposed versions with many metaheuristic algorithms as the native Butterfly Optimization Algorithm (BOA), Grey Wolf Optimizer (GWO), Moth-flame Optimization (MFO), Particle warm Optimization (PSO), Sine Cosine Algorithm (SCA), and Whale Optimization Algorithm (WOA) [54].

The individual search agent is set to 50 and the maximum number of iteration is fixed to 500. The parameters setting of all comparative algorithms is given in Table 2.

**3.1.3 Results & discussion.** In this section, the proposed versions (OBBOA, CLSBOA, and CLSOBBOA) are presented and compared with the original BOA as shown in Table 3. From this table, it has been noticed that the 3rd proposed version called (CLSOBBOA) have achieved the best results in terms of Average/Mean, Best, Worst, and Standard Deviation (SD).

Table 4 shows the comparison of CLSOBBOA (the best proposed version) with other state-of-art metaheuristics algorithm. It's noticed that CLSOBBOA achieve best results and ranked first in almost half of the benchmark functions. Figs 1, 2 and 3 show the convergence curve of these functions. Also, Wilcoxon rank sum [55, 56] test has been performed between CLSOB-BOA and the native BOA as given in Table 5 where the significance level has been considered 5%.

Furthermore, Figs 4 and 5 show the box plot for some functions: unimodal(F1 and F3), multi-modal(F4, F7, F9, F11, F13, and F16), hybrid (F18, F20, F21 and F22), and Composite functions(F25, F27, F28, and F30). It's obvious that CLSOBBOA is more narrow than original BOA and it's super narrow compared with other comparative metaheuristics algorithms.

## 3.2 Engineering problem

In order to evaluate a metaheuristics algorithm, a common approach is testing it on real constrained Engineering problems. These engineering problems have many equality and inequality. In addition, the optimal parameter values of almost engineering problems are unknown. In this subsection, 4 engineering optimization problems are used to test CLSOBBOA. These problems are welded beam engineering design, tension/compression spring, pressure vessel design, and Speed reducer design problem.

**3.2.1 Welded beam design problem.** This engineering problem proposed by Coello in [57] has 4 parameters. These parameters are design thickness of the weld *h*, clamped bar length *l*, bar thickness *b*, and the height of the bar t. The mathematical representation can be expounded in Appendix 6.1. Table 6 shows the results of CLSOBBOA compared with Animal

**Table 3. The comparison results of all algorithms over 30 functions.**

| F | Algorithm | Best | Worst | Mean | SD |
|---|-----------|------|-------|------|-----|
| F1 | BOA | 3.5971e+07 | 3.1810e+08 | 1.0080e+06 | 1.2667e+5 |
| | OBBOA | 1.6723e+07 | 2.3640e+08 | 6.7445e+07 | 4.7897e+07 |
| | CLSBOA | 5.7586e+07 | 7.2621e+08 | 1.2429e+08 | 1.4759e+08 |
| | CLSOBBOA | 9.5454e+04 | 1.9320e+07 | 8.0108e+07 | 7.2858e+07 |
| F2 | BOA | 2.6574e+09 | 1.0043e+10 | 4.4261e+09 | 2.5605e+09 |
| | OBBOA | 7.1006e+08 | 8.8216e+09 | 3.3621e+09 | 1.8186e+09 |
| | CLSBOA | 2.2787e+09 | 9.2016e+09 | 4.1975e+09 | 2.0389e+09 |
| | CLSOBBOA | 6.6739e+03 | 6.3838e+07 | 3.2066e+05 | 4.5402e+3 |
| F3 | BOA | 1.2913e+04 | 1.8349e+04 | 1.4306e+04 | 2.5048e+03 |
| | OBBOA | 7.2557e+03 | 1.7454e+04 | 1.2592e+04 | 2.7249e+03 |
| | CLSBOA | 1.1739e+04 | 1.7093e+04 | 1.3854e+04 | 2.5865e+03 |
| | CLSOBBOA | 8.5819e+03 | 1.5012e+04 | 1.1610e+04 | 1.7285e+3 |
| F4 | BOA | 2.0912e+03 | 3.8529e+03 | 2.6292e+03 | 5.6597e+02 |
| | OBBOA | 1.2404e+03 | 4.4836e+03 | 2.3235e+03 | 8.2272e+02 |
| | CLSBOA | 1.9510e+03 | 5.3563e+03 | 2.7072e+03 | 1.0699e+03 |
| | CLSOBBOA | 4.2516e+2 | 2.7130e+03 | 8.8079e+02 | 13.310 |
| F5 | BOA | 5.2042e+02 | 5.2066e+02 | 5.2049e+02 | 0.1050 |
| | OBBOA | 5.2036e+02 | 5.2061e+02 | 5.2047e+02 | 0.0786 |
| | CLSBOA | 5.2032e+02 | 5.2052e+02 | 5.2038e+02 | 0.0775 |
| | CLSOBBOA | 5.2028e+02 | 5.2064e+02 | 5.2040e+02 | 0.0565 |
| F6 | BOA | 6.0708e+02 | 6.0956e+02 | 6.0832e+02 | 1.0965 |
| | OBBOA | 6.0725e+02 | 6.0911e+02 | 6.0840e+02 | 0.5863 |
| | CLSBOA | 6.0770e+02 | 6.1002e+02 | 6.0850e+02 | 0.9281 |
| | CLSOBBOA | 6.0190e+02 | 6.1009e+02 | 6.0843e+02 | 0.577 |
| F7 | BOA | 8.0304e+02 | 9.5780e+02 | 8.7396e+02 | 62.3545 |
| | OBBOA | 7.6548e+02 | 9.7498e+02 | 8.5850e+02 | 56.6647 |
| | CLSBOA | 8.1979e+02 | 8.8123e+02 | 8.4222e+02 | 36.6645 |
| | CLSOBBOA | 7.0012e+02 | 8.8830e+02 | 7.3922e+02 | 0.06032 |
| F8 | BOA | 8.6394e+02 | 8.8581e+02 | 8.7199e+02 | 10.1535 |
| | OBBOA | 8.5749e+02 | 8.9292e+02 | 8.7059e+02 | 9.0216 |
| | CLSBOA | 8.5810e+02 | 8.9173e+02 | 8.6610e+02 | 11.4144 |
| | CLSOBBOA | 8.0436e+2 | 8.8665e+02 | 8.3193e+02 | 2.56771 |
| F9 | BOA | 9.6165e+02 | 9.7920e+02 | 9.6592e+02 | 9.0121 |
| | OBBOA | 9.4129e+02 | 9.8231e+02 | 9.6468e+02 | 10.3115 |
| | CLSBOA | 9.5419e+02 | 9.8318e+02 | 9.6244e+02 | 11.6616 |
| | CLSOBBOA | 9.5529e+02 | 9.7704e+02 | 9.6255e+02 | 6.2637 |
| F10 | BOA | 2.5486e+03 | 3.0370e+03 | 2.6438e+03 | 1.8702e+02 |
| | OBBOA | 2.2832e+03 | 3.0108e+03 | 2.5974e+03 | 1.9311e+02 |
| | CLSBOA | 2.5452e+03 | 3.0265e+03 | 2.6622e+03 | 2.1390e+02 |
| | CLSOBBOA | 1.1924e+3 | 3.0311e+03 | 1.6173e+03 | 1.4853e+02 |
| F11 | BOA | 2.6618e+03 | 3.1056e+03 | 2.7892e+03 | 1.6215e+02 |
| | OBBOA | 2.2947e+03 | 3.1046e+03 | 2.7424e+03 | 2.1860e+02 |
| | CLSBOA | 2.6374e+03 | 3.2235e+03 | 2.7841e+03 | 2.2713e+02 |
| | CLSOBBOA | 1.7170e+03 | 2.8534e+03 | 2.7774e+03 | 1.6215e+2 |

(*Continued*)

**Table 3.** (Continued)

| F | Algorithm | Best | Worst | Mean | SD |
|---|---|---|---|---|---|
| F12 | BOA | 1.2017e+03 | 1.2023e+03 | 1.2019e+03 | 0.2708 |
| | OBBOA | 1.2011e+03 | 1.2022e+03 | 1.2017e+03 | 0.3247 |
| | CLSBOA | 1.2015e+03 | 1.2021e+03 | 1.2017e+03 | 0.2306 |
| | CLSOBBOA | 1.2009e+03 | 1.2019e+03 | 1.2015e+03 | 0.1381 |
| F13 | BOA | 1.3039e+03 | 1.3054e+03 | 1.3045e+03 | 0.6891 |
| | OBBOA | 1.3032e+03 | 1.3052e+03 | 1.3041e+03 | 0.5280 |
| | CLSBOA | 1.3037e+03 | 1.3062e+03 | 1.3044e+03 | 1.0008 |
| | CLSOBBOA | 1.3001e+03 | 1.3053e+03 | 1.3022e+03 | 0.05490 |
| F14 | BOA | 1.4282e+03 | 1.4504e+03 | 1.4354e+03 | 9.3979 |
| | OBBOA | 1.4245e+03 | 1.4490e+03 | 1.4379e+03 | 6.8308 |
| | CLSBOA | 1.4333e+03 | 1.4541e+03 | 1.4373e+03 | 8.0703 |
| | CLSOBBOA | 1.4002e+03 | 1.4465e+03 | 1.4320e+03 | 0.1292 |
| F15 | BOA | 2.8689e+03 | | 6.500e+03 | 5.0786e+03 |
| | OBBOA | 1.9296e+03 | 1.3932e+04 | 4.4538e+03 | 3.4833e+03 |
| | CLSBOA | 3.2042e+03 | 3.1369e+04 | 7.7426e+03 | 7.2418e+03 |
| | CLSOBBOA | 1.5024e+03 | 1.0747e+04 | 4.7610e+03 | 1.0485e+02 |
| F16 | BOA | 1.6035e+03 | 1.6038e+03 | 1.6036e+03 | 0.1655 |
| | OBBOA | 1.6033e+03 | 1.6038e+03 | 1.6036e+03 | 0.1570 |
| | CLSBOA | 1.6033e+03 | 1.6037e+03 | 1.6035e+03 | 0.2070 |
| | CLSOBBOA | 1.6033e+3 | 1.6038e+03 | 1.6035e+03 | 0.0598 |
| F17 | BOA | 2.3517e+05 | 5.2617e+05 | 3.3745e+05 | 1.1954e+05 |
| | OBBOA | 7.6198e+04 | 5.1763e+05 | 2.2619e+05 | 1.2488e+05 |
| | CLSBOA | 3.1117e+05 | 6.5610e+05 | 4.0887e+05 | 1.5533e+05 |
| | CLSOBBOA | 4.8701e+04 | 7.0552e+05 | 8.5776e+04 | 2.5502e+04 |
| F18 | BOA | 1.7587e+04 | 4.8365e+06 | 3.0925e+05 | 1.0695e+06 |
| | OBBOA | 1.0590e+04 | 1.7808e+06 | 1.2400e+05 | 3.9117e+05 |
| | CLSBOA | 1.3717e+04 | 1.6055e+06 | 1.3132e+05 | 3.6880e+05 |
| | CLSOBBOA | 7.9930e+3 | 1.3861e+05 | 4.6620e+04 | 3.3053 |
| F19 | BOA | 1.9279e+03 | 1.9786e+03 | 1.9389e+03 | 18.0575 |
| | OBBOA | 1.9071e+03 | 1.9772e+03 | 1.9268e+03 | 19.6003 |
| | CLSBOA | 1.9265e+03 | 2.0442e+03 | 1.9461e+03 | 29.3827 |
| | CLSOBBOA | 1.9026e+3 | 1.9512e+03 | 1.9250e+03 | 2.9060 |
| F20 | BOA | 7.6669e+03 | 8.6363e+04 | 2.0606e+04 | 1.98136e+04 |
| | OBBOA | 2.2118e+03 | 3.0375e+04 | 1.3241e+04 | 8.30101e+03 |
| | CLSBOA | 9.5177e+03 | 7.5429e+04 | 1.8474e+04 | 1.64929e+04 |
| | CLSOBBOA | 5.1116e+03 | 3.6863e+04 | 1.0850e+04 | 8.91533e+03 |
| F21 | BOA | 4.5448e+04 | 1.6143e+06 | 3.2084e+05 | 4.02760e+05 |
| | OBBOA | 1.7469e+04 | 9.8848e+05 | 1.5497e+05 | 2.17338e+05 |
| | CLSBOA | 2.7288e+04 | 6.7756e+05 | 1.8627e+05 | 2.16049e+05 |
| | CLSOBBOA | 3.6120e+03 | 4.0706e+04 | 1.3284e+04 | 1.5581e+03 |
| F22 | BOA | 2.4161e+03 | 2.5767e+03 | 2.4490e+03 | 61.2592 |
| | OBBOA | 2.2804e+03 | 2.4836e+03 | 2.3877e+03 | 57.6424 |
| | CLSBOA | 2.3370e+03 | 2.7499e+03 | 2.4365e+03 | 1.05537e+02 |
| | CLSOBBOA | 2.230e+03 | 2.4935e+03 | 2.3890e+03 | 18.5703 |

(*Continued*)

**Table 3.** (Continued)

| F | Algorithm | Best | Worst | Mean | SD |
|---|---|---|---|---|---|
| F23 | BOA | 2.5000e+03 | 2500 | 2500 | |
| | OBBOA | 2.5000e+03 | 2.5000e+03 | 2500 | 4.1730e-13 |
| | CLSBOA | 2.5000e+03 | 2500 | 2500 | |
| | CLSOBBOA | 2.5000e+03 | 2.5000e+03 | 2500 | 4.1730e-13 |
| F24 | BOA | 2.5795e+03 | 2600 | 2.5918e+03 | 12.1467 |
| | OBBOA | 2.5544e+03 | 2600 | 2.5877e+03 | 14.9424 |
| | CLSBOA | 2.5927e+03 | 2600 | 2.5968e+03 | 5.6880 |
| | CLSOBBOA | 2.5592e+033 | 2600 | 2.5907e+03 | 8.6636 |
| F25 | BOA | 2700 | 2700 | 2.6982e+03 | 5.4556 |
| | OBBOA | 2.6822e+03 | 2.7000e+03 | 2.6978e+03 | 5.4111 |
| | CLSBOA | 2700 | 2700 | 2.6990e+03 | 2.9984 |
| | CLSOBBOA | 2.682e+03 | 2.7000e+03 | 2.6998e+03 | 5.45561 |
| F26 | BOA | 2.7023e+03 | 2.7067e+03 | 2.7034e+03 | 1.7518 |
| | OBBOA | 2.7003e+03 | 2.7033e+03 | 2.7016e+03 | 0.9510 |
| | CLSBOA | 2.7023e+03 | 2.7249e+03 | 2.7043e+03 | 5.0978 |
| | CLSOBBOA | 2.7008e+03 | 2.7033e+03 | 2.7018e+03 | 0.0909 |
| F27 | BOA | 2.8612e+03 | 3.2305e+03 | 3.0001e+03 | 1.5009e+02 |
| | OBBOA | 2.7465e+03 | 3.1371e+03 | 2.9313e+03 | 1.4431e+02 |
| | CLSBOA | 2.7710e+03 | 2.9000e+03 | 2.8521e+03 | 69.0391 |
| | CLSOBBOA | 2.7480e+03 | 2.9000e+03 | 2.8568e+03 | 0.469+e02 |
| F28 | BOA | 3000 | 3.5324e+03 | 3.1976e+03 | 2.08291e+02 |
| | OBBOA | 3.3249e+03 | 3.6655e+03 | 3.5018e+03 | 1.02142e+02 |
| | CLSBOA | 3.0000e+03 | 3.0000e+03 | 3.0000e+03 | 0.0054 |
| | CLSOBBOA | 3.0000e+03 | 3.0000e+03 | 3.0000e+03 | 1.790e-4 |
| F29 | BOA | 3100 | 1.1511e+05 | 2.4659e+04 | 3.3954e+04 |
| | OBBOA | 3100 | 1.0710e+06 | 2.7354e+05 | 3.7340e+05 |
| | CLSBOA | 3100 | 3.5565e+04 | 4.7232e+03 | 7.2595e+03 |
| | CLSOBBOA | 3100 | 5.8748e+05 | 3.7343e+04 | 1.6260e+03 |
| F30 | BOA | 5.9971e+03 | 2.6134e+04 | 1.0799e+04 | 6.7561e+03 |
| | OBBOA | 3.2000e+03 | 3.7058e+04 | 1.2980e+04 | 8.5345e+03 |
| | CLSBOA | 3200 | 2.4155e+04 | 8.0275e+03 | 6.3396e+03 |
| | CLSOBBOA | 4.1627e+03 | 5.8775e+04 | 1.1109e+04 | 8.4793e+02 |

Migration Optimization (AMO) [58], Water cycle algorithm (WCA) [59], Lightning search algorithm (LSA) [60], Symbiotic organisms search (SOS) [61], and Grey Wolf Optimizer (GWO) [62].

**3.2.2 Tension/Compression spring.** The second engineering constrained problem is called Tension/Compression spring proposed by Arora [63]. The main goal of this problem is to minimize the weight of design spring by find the optimal values for the 3 parameters: the diameter of the wire $d$, the mean diameter of the coil $D$ and the active coil numbers $N$. Also, Appendix 6.2 gives its mathematical definition. Table 7 compares the results of CLSOBBOA algorithm with WCA, ABC [64], TLBO [65], and SOS.

**3.2.3 Pressure vessel design.** One of the most famous engineering problem is the pressure vessel design introduced by Kannan and Kramer in [66] which aims to minimize the cost of materials, welding, and forming This problem has 4 parameters: the thickness $T_s$, head's thickness $T_h$, the inner radius R, and cylindrical length $L$. Mathematical definition of this problem

**Table 4. The comparison results of all algorithms over 30 functions.**

| | F1 | | F2 | | F3 | |
|---|---|---|---|---|---|---|
| | Avg | Std | Avg | Std | Avg | Std |
| CLSOBBOA | **9.5454e+4** | **1.2667e+5** | 6.6739e+3 | 4.5402e+3 | 8.5819e+03 | **1.7285e+3** |
| BOA | 1.0080e+08 | 5.2571e+07 | 4.4261e+09 | 2.5605e+09 | 1.4306e+04 | 2.5048e+03 |
| GWO | 9.5526e+06 | 5.0611e+06 | 9.0682e+07 | 2.4562e+08 | 1.3114e+04 | 9.0242e+03 |
| MFO | 3.5572e+06 | 7.2399e+06 | 1.1676e+09 | 2.2771e+08 | 1.9628e+04 | 1.5038e+04 |
| PSO | 2.5249e+07 | 8.0505e+06 | **5.0614e+3** | **2.3787e+3** | **5.2453e+3** | 3.9510e+3 |
| SCA | 1.2039e+07 | 5.1675e+06 | 9.3345e+08 | 4.7255e+08 | 1.1411e+04 | 8.8356e+03 |
| WOA | 1.1876e+07 | 8.1442e+06 | 2.0966e+07 | 1.2829e+07 | 5.9297e+04 | 3.9137e+04 |
| | **F4** | | **F5** | | **F6** | |
| | Avg | Std | Avg | Std | Avg | Std |
| CLSOBBOA | **4.2516e+2** | **13.310** | 5.2028e+02 | **0.0565** | **6.019e+2** | **0.577** |
| BOA | 2.6292e+03 | 5.6597e+02 | 5.2049e+02 | 0.1050 | 6.0832e+02 | 1.0965 |
| GWO | 4.3397e+02 | 5.9297 | 5.2044e+02 | 0.1227 | 6.0253e+02 | 1.0790 |
| MFO | 4.2751e+02 | 1.3855e+02 | **5.2012e+2** | 0.1329 | 6.0456e+02 | 1.7994 |
| PSO | 1.1304e+03 | 20.343 | 5.2040e+02 | 0.1073 | 6.0722e+02 | 1.0849 |
| SCA | 4.9472e+02 | 32.142 | 5.2048e+02 | 0.1230 | 6.0762e+02 | 1.4810 |
| WOA | 4.5614e+02 | 34.900 | 5.2024e+02 | 0.1089 | 6.0854e+02 | 1.5315 |
| | **F7** | | **F8** | | **F9** | |
| | Avg | Std | Avg | Std | Avg | Std |
| CLSOBBOA | **7.0012e+2** | **0.06032** | **8.0436e+2** | **2.56771** | 9.5529e+02 | 6.2637 |
| BOA | 8.7396e+02 | 62.3545 | 8.7199e+02 | 10.1535 | 9.6592e+02 | 9.0121 |
| GWO | 7.0123e+02 | 0.77348 | 8.1427e+02 | 6.45195 | 9.1895e+02 | 7.9132 |
| MFO | 8.0137e+02 | 16.4145 | 8.2414e+02 | 10.6745 | 9.3011e+02 | 12.381 |
| PSO | 7.0097e+02 | 2.05917 | 8.5830e+02 | 6.3450 | **9.1282e+2** | **4.4846** |
| SCA | 7.1329e+02 | 4.26272 | 8.4631e+02 | 11.2488 | 9.5284e+02 | 9.6501 |
| WOA | 7.0165e+02 | 0.50518 | 8.5151e+02 | 20.8518 | 9.4555e+02 | 20.916 |
| | **F10** | | **F11** | | **F12** | |
| | Avg | Std | Avg | Std | Avg | Std |
| CLSOBBOA | **1.1924e+3** | 1.4853e+02 | **1.7170e+3** | **1.6215e+2** | **1.2009e+3** | **0.1381** |
| BOA | 2.6438e+03 | 1.8702e+02 | 2.7892e+03 | 2.9413e+02 | 1.2019e+03 | 0.2708 |
| GWO | 1.4089e+03 | 1.9919e+02 | 2.3330e+03 | 1.6815e+02 | 1.2012e+03 | 0.6264 |
| MFO | 1.5960e+03 | 2.5578e+02 | 2.0165e+03 | 3.0732e+02 | 1.2003e+03 | 0.2121 |
| PSO | 2.3420e+03 | **1.2159e+2** | 1.7719e+03 | 3.6301e+02 | 1.2013e+03 | 0.4253 |
| SCA | 2.0964e+03 | 2.4770e+02 | 2.5883e+03 | 1.9564e+02 | 1.2015e+03 | 0.3095 |
| WOA | 1.6769e+03 | 3.5197e+02 | 2.2302e+03 | 3.3642e+02 | 1.2012e+03 | 0.3191 |
| | **F13** | | **F14** | | **F15** | |
| | Avg | Std | Avg | Std | Avg | Std |
| CLSOBBOA | **1.3001e+3** | **0.05490** | **1.4002e+3** | 0.1292 | 1.5024e+3 | 1.04857 |
| BOA | 1.3045e+03 | 0.6891 | 1.4354e+03 | 9.3979 | 6.5005e+03 | 5.07867e+03 |
| GWO | 1.3002e+03 | 0.06616 | 1.4004e+03 | 0.1898 | 1.8759e+03 | 2.0916e+02 |
| MFO | 1.3003e+03 | 0.16621 | 1.4007e+03 | 1.0447 | 1.5041e+03 | 10.8810 |
| PSO | 1.3034e+03 | 0.24070 | **1.4002e+3** | **0.0585** | **1.5014e+3** | **0.75305** |
| SCA | 1.3007e+03 | 0.17544 | 1.4015e+03 | 0.6502 | 1.5110e+03 | 3.99372 |
| WOA | 1.3004e+03 | 0.18968 | 1.4243e+03 | 5.1756 | 1.5086e+03 | 6.54693 |
| | **F16** | | **F17** | | **F18** | |
| | Avg | Std | Avg | Std | Avg | Std |
| CLSOBBOA | **1.6033e+3** | **0.0598** | 4.8701e+04 | 2.5502e+04 | **7.9930e+3** | **3.3053e+3** |

*(Continued)*

**Table 4.** (Continued)

| | | | | | | |
|---|---|---|---|---|---|---|
| BOA | 1.6036e+03 | 0.1655 | 3.3745e+05 | 1.1954e+05 | 3.0925e+05 | 1.06959e+06 |
| GWO | 1.6028e+03 | 0.3827 | 7.0802e+04 | 1.5951e+05 | 1.3989e+04 | 1.06251e+04 |
| MFO | **1.6033e+3** | 0.4842 | 1.9565e+05 | 3.3550e+05 | 2.2320e+04 | 1.50917e+04 |
| PSO | 1.6028e+03 | 0.4233 | **1.2951e+4** | **2.0212e+4** | 9.1989e+03 | 1.06636e+04 |
| SCA | 1.6035e+03 | 0.2271 | 6.4658e+04 | 1.5044e+05 | 3.0945e+04 | 2.02963e+04 |
| WOA | 1.6036e+03 | 0.3439 | 1.9715e+05 | 3.3453e+05 | 1.7006e+04 | 1.33556e+04 |
| | **F19** | | **F20** | | **F21** | |
| | Avg | Std | Avg | Std | Avg | Std |
| CLSOBBOA | **1.9026e+3** | 0.9060 | **5.1116e+3** | **1.5581e+3** | **3.6120e+3** | **2.0764e+3** |
| BOA | 1.9389e+03 | 18.0575 | 2.0606e+04 | 1.9813e+04 | 3.2084e+05 | 4.0276e+05 |
| GWO | 1.9118e+03 | 2.1492 | 1.0008e+04 | 5.9314e+03 | 1.2467e+04 | 6.3934e+03 |
| MFO | 1.9029e+03 | **0.8251** | 1.5952e+04 | 1.9123e+04 | 1.2523e+04 | 1.1755e+04 |
| PSO | 1.9027e+03 | 1.3841 | 8.2582e+03 | 6.5637e+03 | 2.0928e+04 | 1.2878e+04 |
| SCA | 1.9061e+03 | 1.0124 | 8.7350e+03 | 5.4998e+03 | 1.8935e+04 | 1.0503e+04 |
| WOA | 1.9070e+03 | 1.9011 | 1.4986e+04 | 8.6110e+03 | 2.2521e+05 | 3.2664e+05 |
| | **F22** | | **F23** | | **F24** | |
| | Avg | Std | Avg | Std | Avg | Std |
| CLSOBBOA | **2.230e+3** | **18.5703** | 2500 | 9.53030 | 2.5592e+03 | 8.6636 |
| BOA | 2.4490e+03 | 61.2592 | 2500 | 8.43650 | 2.5918e+03 | 12.1467 |
| GWO | 2.3164e+03 | 61.7699 | 2.6324e+03 | 3.01732 | 2.5271e+03 | 15.6449 |
| MFO | 2.3047e+03 | 75.2092 | 2.6347e+03 | 6.72352 | 2.5443e+03 | 16.1036 |
| PSO | 2.3058e+03 | 38.2640 | 2.6294e+03 | **1.922e-07** | **2.522e+03** | **6.555** |
| SCA | 2.2910e+03 | 28.2535 | 2.6497e+03 | 8.06636 | 2.5582e+03 | 9.27922 |
| WOA | 2.3114e+03 | 81.5165 | 2.6191e+03 | 51.8188 | 2.5903e+03 | 21.0583 |
| | **F25** | | **F26** | | **F27** | |
| | Avg | Std | Avg | Std | Avg | Std |
| CLSOBBOA | **2.682e+03** | **5.45561** | **2.7001e+03** | 0.0909 | **2.748e+03** | **0.469+e02** |
| BOA | 2.6982e+03 | 9.4997 | 2.7034e+03 | 1.7518 | 3.0001e+03 | 1.5009e+02 |
| GWO | 2.6953e+03 | 17.1437 | **2.7001e+03** | **0.0563** | 3.0280e+03 | 1.1592e+02 |
| MFO | 2.6991e+03 | 17.1635 | 2.7002e+03 | 0.1785 | 3.0685e+03 | 1.2750e+02 |
| PSO | 2.6918e+03 | 34.4413 | **2.7001e+03** | 0.0736 | 2.9463e+03 | 1.6596e+02 |
| SCA | 2.7004e+03 | 7.30156 | 2.7008e+03 | 0.1900 | 3.0131e+03 | 1.6786e+02 |
| WOA | 2.6968e+03 | 9.23225 | 2.7004e+03 | 0.1786 | 3.0791e+03 | 2.0168e+02 |
| | **F28** | | **F29** | | **F30** | |
| | Avg | Std | Avg | Std | Avg | Std |
| CLSOBBOA | **3.000e+03** | **1.790e-4** | **3.100e+03** | 1.6260e+03 | 4.1627e+03 | 8.4793e+02 |
| BOA | 3.1976e+03 | 2.0829e+02 | 2.4659e+04 | 3.3954e+04 | 1.0799e+04 | 6.7561e+03 |
| GWO | 3.2956e+03 | 87.2767 | 8.5841e+05 | 1.0925e+06 | 4.4923e+03 | 7.4357e+02 |
| MFO | 3.1988e+03 | 36.6856 | 3.8029e+03 | **4.636e+02** | **3.795e+03** | **2.893e+02** |
| PSO | 3.2615e+03 | 65.2974 | 8.0351e+05 | 1.6493e+06 | 3.9944e+03 | 3.8509e+02 |
| SCA | 3.2828e+03 | 53.7555 | 1.0608e+04 | 6.0870e+03 | 5.0231e+03 | 1.0682e+03 |
| WOA | 3.4616e+03 | 1.7564e+02 | 6.3032e+05 | 1.0588e+06 | 6.0717e+03 | 1.5832e+03 |

is shown in Appendix 6.3. Results of CLSOBBOA compared to other state-of-art algorithms LSA, SOS, ABC and GWO is shown in Table 8.

**3.2.4 Speed reducer design problem.** The last engineering problem introduced in this section is the speed reducer problem The objective of the function ids to fond the best

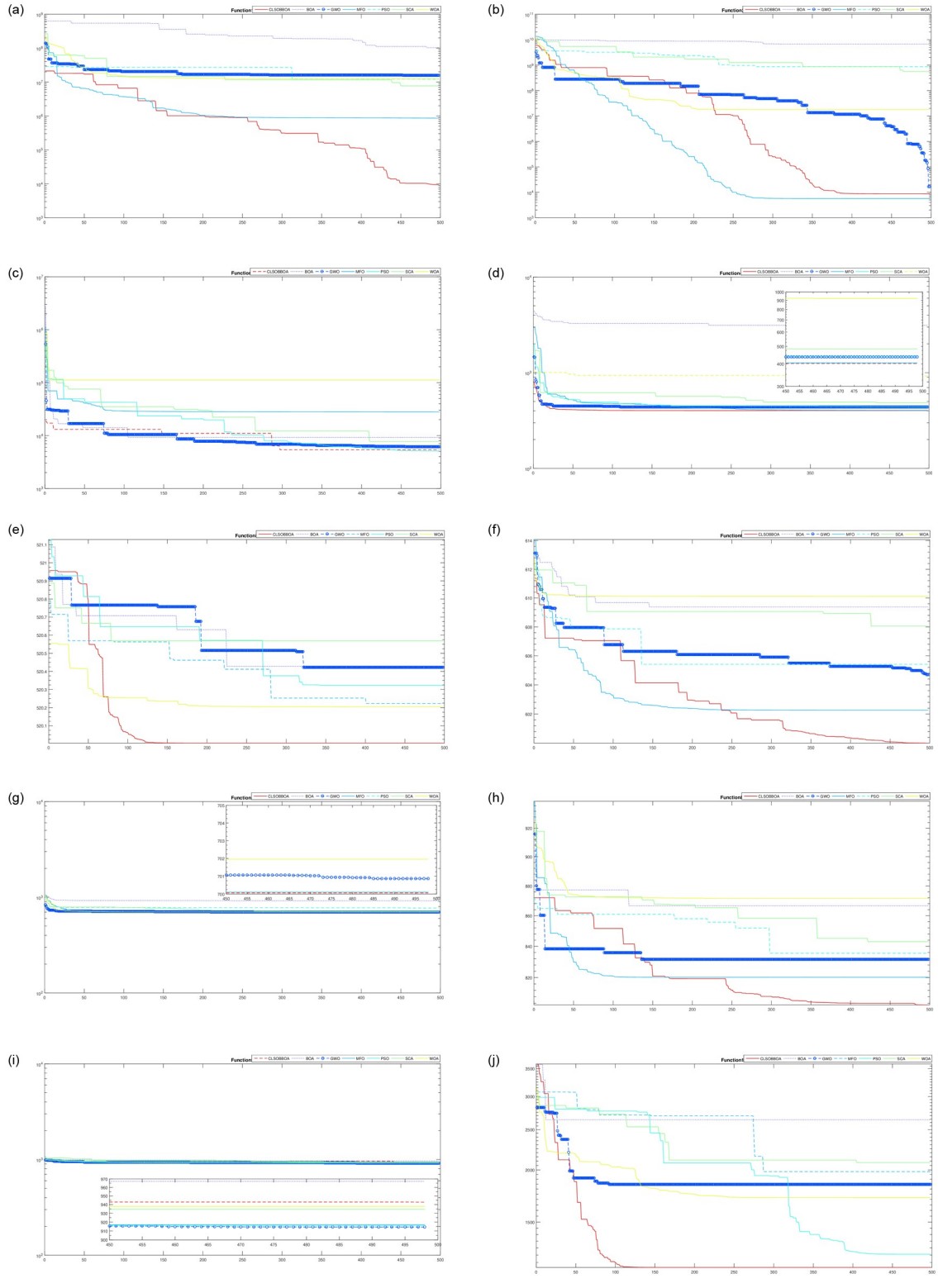

**Fig 1. Convergence curve for all algorithms from F1–F10.**

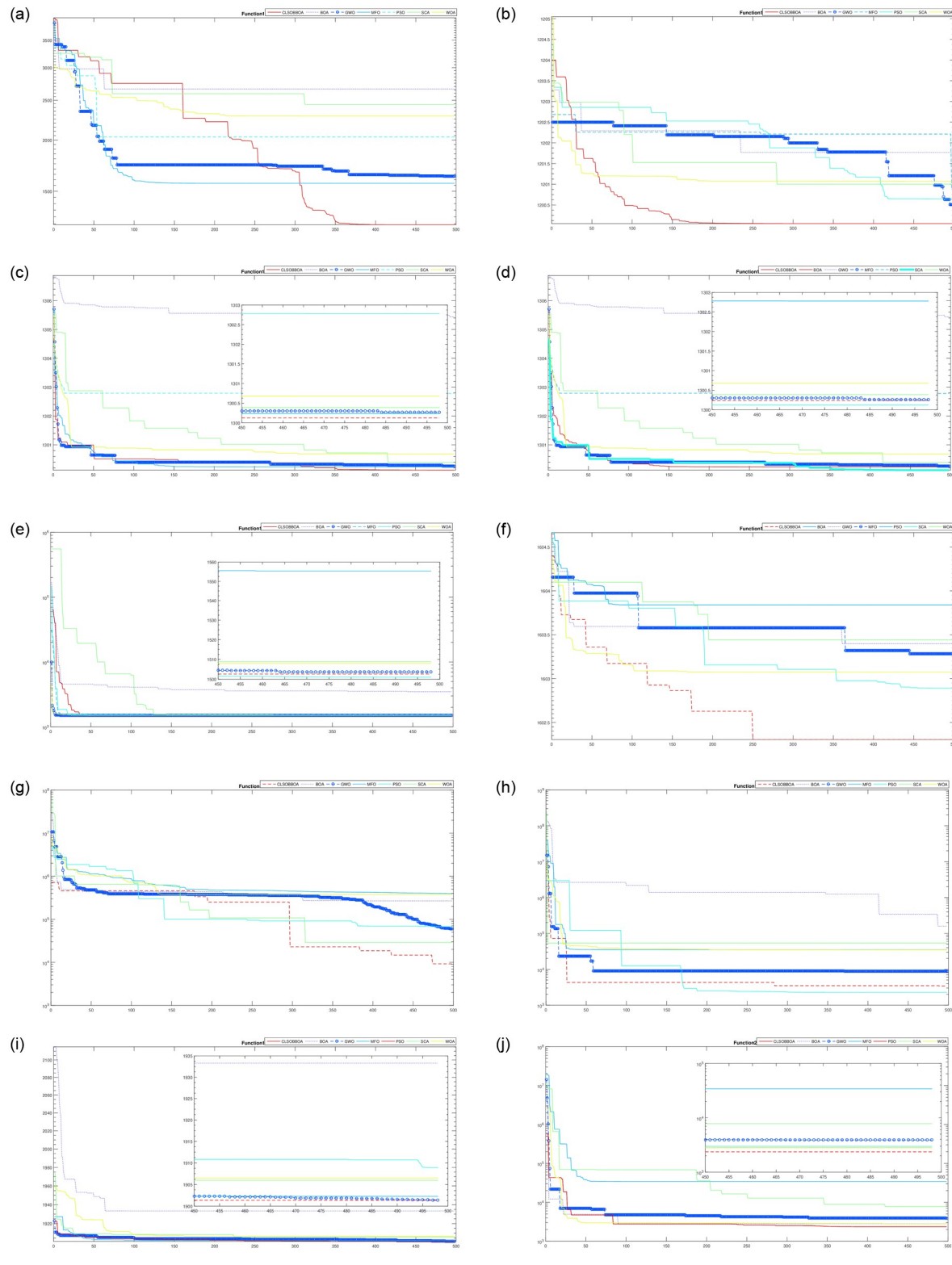

**Fig 2. Convergence curve for all algorithms from F10–F20.**

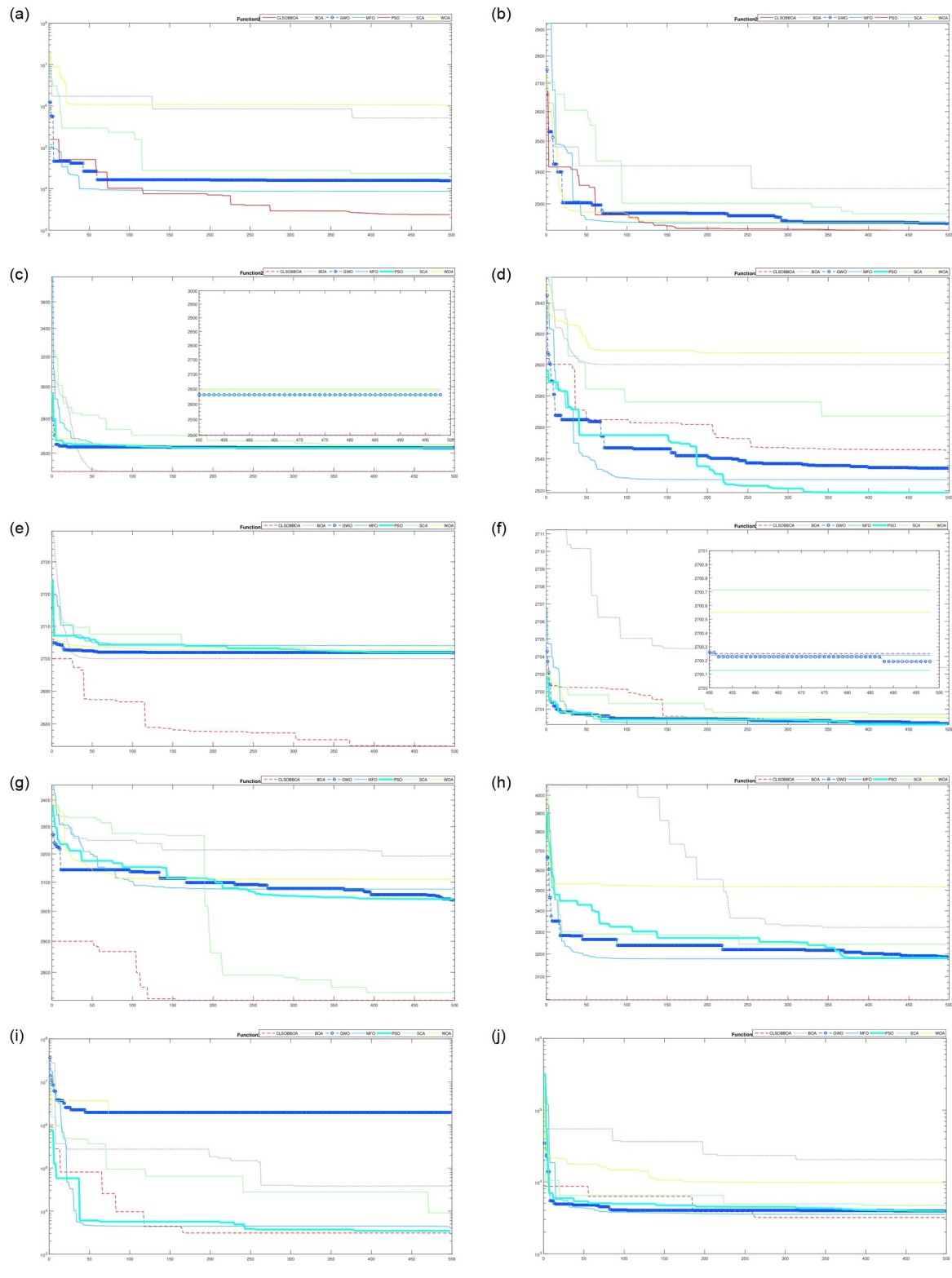

**Fig 3. Convergence curve for all algorithms from F20–F30.**

**Table 5. Results of Wilcoxon signed rank test.**

| Fun. | p-value | Decision | Fun. | p-value | Decision |
|------|---------|----------|------|---------|----------|
| F1 | 6.4e-10 | + | F2 | 2.7e-8 | + |
| F3 | 4.4e-6 | + | F4 | 2.4e-5 | + |
| F5 | 3.3e-5 | + | F6 | 7.3e-6 | + |
| F7 | 4.8e-5 | + | F8 | 6.2e-6 | + |
| F9 | 4.3e-4 | + | F10 | 4.3e-8 | + |
| F11 | 5.1e-6 | + | F12 | 2.4e-6 | + |
| F13 | 6.9e-4 | + | F14 | 3.7e-5 | + |
| F15 | 2.4e-3 | + | F16 | 2.2e-4 | + |
| F17 | 3.5e-4 | + | F18 | 4.8e-5 | + |
| F19 | 1.3e-6 | + | F20 | 3.8e-5 | + |
| F21 | 4.1e-6 | + | F22 | 6.4e-6 | + |
| F23 | 6.7e-4 | + | F24 | 4.7e-5 | + |
| F25 | 2.7e-3 | + | F26 | 4.2e-4 | + |
| F27 | 2.5e-4 | + | F28 | 4.6e-5 | + |
| F29 | 3.3e-6 | + | F30 | 3.8e-5 | + |

parameter which are face weight, teeth on pinion number, teeth module, shaft length 1 between bearings and the shaft length 2 between bearings. The Mathematical representation is shown in Appendix 6.4. Table 9 compare the results of CLSOBBOA with GWO, AMO, WCA, and SOS.

## 3.3 CLSOBBOA in Feature Selection (FS)

In this subsection CLSOBBOA is used in order to solve FS using 5 different datasets.

**3.3.1 CLSOBBOA architecture of FS.** To be able to solve feature selection (FS), we regard it as a binary optimization since the solutions are limited to 0, 1 where "0" refers to the corresponding attribute hasn't be selected whereas "1" is its contrary. To convert continous solution to binary one, a transfer function is needed. In this paper, we use sigmoid function as shown in the following equation

$$y^k = \frac{1}{1 + e^{-x_i^k(t)}} \qquad (7)$$

where $x_i^k$ refers to the position of i-th agent at dimension k.

The output from the previous equation is still continuous and to have binary-valued one, the following stochastic equation is used

$$x_i^k = \begin{cases} 1 & if rand < S(x_i^k(t+1)) \\ 0 & otherwise \end{cases} \qquad (8)$$

FS fitness function is finding the small feature number and achieving the highest accuracy. So the FS fitness equation is as follows:

$$Fitness = \alpha\gamma(D) + \beta\frac{|R|}{|C|} \qquad (9)$$

where $\gamma(D)$ refers to error rate, $C$ is the features total number, $R$ is the length-size of selected features. $\alpha$ and $\beta$ can be calculated as $\alpha \in [0, 1]$ and $\beta = 1 - \alpha$

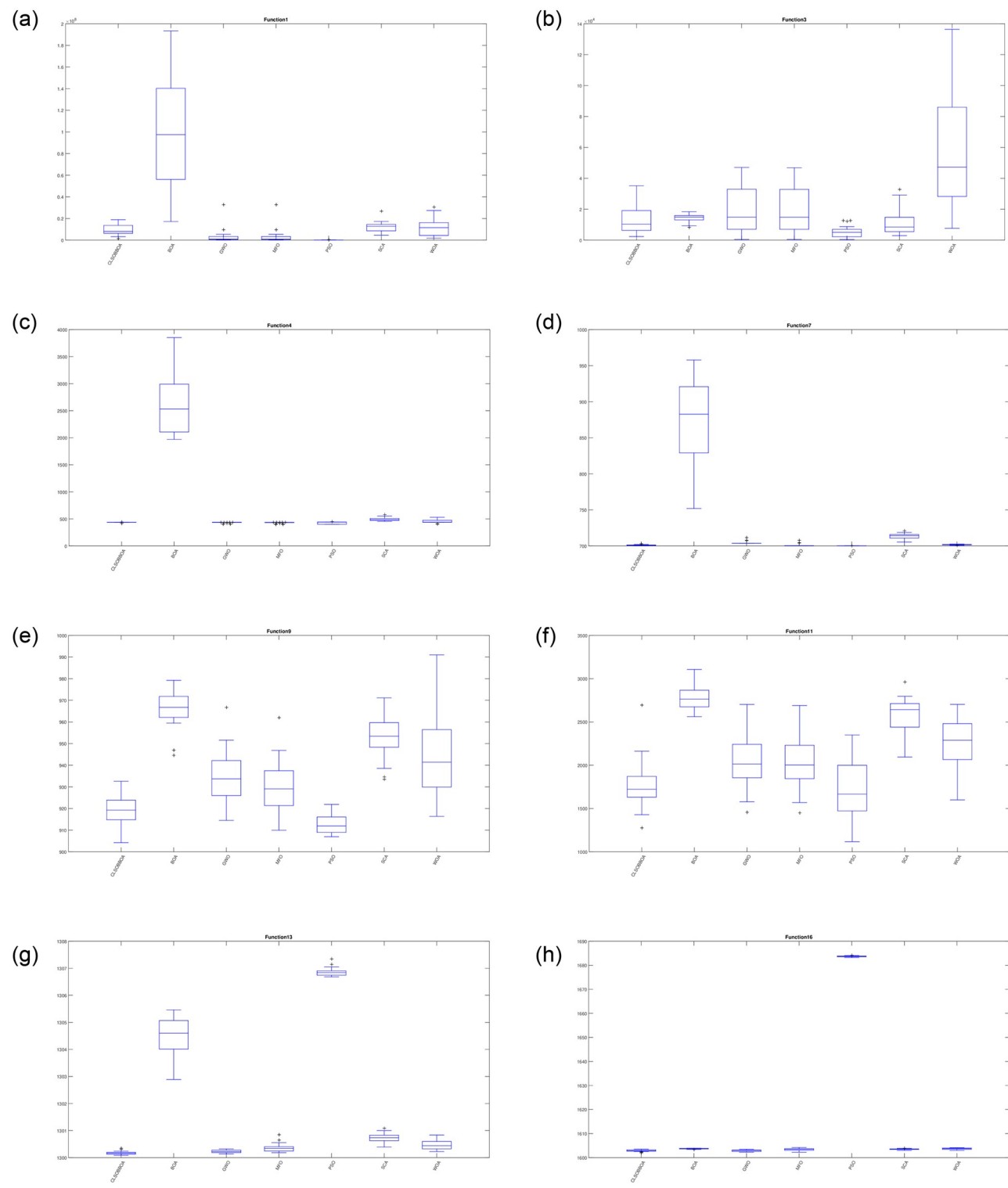

**Fig 4. Box plot for some unimodal and multi modal functions.**

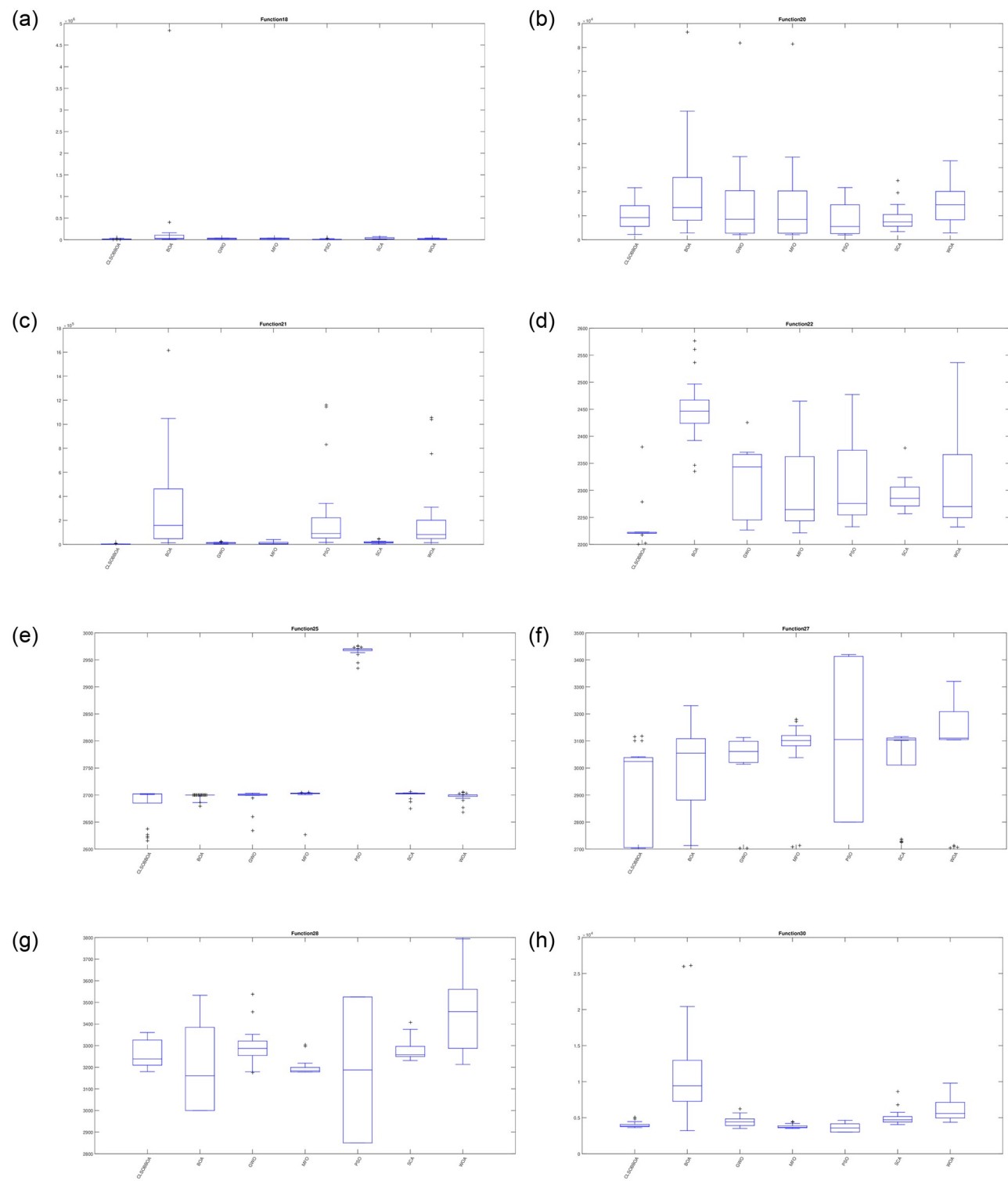

**Fig 5. Box plot for some hybrid and composite functions.**

**Table 6. Optimization results for welded beam design problem.**

| Algorithm | Optimization results | | | | Cost |
|---|---|---|---|---|---|
| | h | l | t | b | |
| CLSOBBOA | 0.205729 | 3.470488 | 9.036622 | 0.205729 | 1.724852 |
| AMO | 0.223 960 | 3.591 024 | 8.834 515 | 0.223 960 | 1.873 459 |
| WCA | 0.205 730 | 3.470 489 | 9.036 624 | 0.205 730 | 1.724 852 315 |
| LSA | 0.205 730 | 3.470 488 | 9.036 623 | 0.205 730 | 1.724 852 526 |
| SOS | 0.205 730 | 3.470 745 | 9.036 354 | 0.205 744 | 1.724 953 103 |
| GWO | 0.205 587 | 3.475 084 | 9.035 006 | 0.205 808 | 1.725 571 417 |

**Table 7. Optimization results for the tension/compression design problem.**

| Algorithm | Optimization results | | | Cost |
|---|---|---|---|---|
| | d | D | N | |
| CLSOBBOA | 0.051 688 | 0.356 715 | 11.289 108 | 0.012 665 |
| WCA | 0.051 773 | 0.358 734 | 11.171 709 | 0.012 665 |
| ABC | 0.052 717 | 0.381 929 | 9.951 875 | 0.012 685 948 |
| TLBO | 0.051 790 | 0.359 142 | 11.148 539 | 0.012 665 851 |
| SOS | 0.051 808 | 0.359 577 | 11.125 | 0.012 667 638 |

**3.3.2 Experimental setup & results.** Here, 5 different datasets from UCI have been used to evalute the CLSOBBOA performance in solving FS problem. The details of each dataset can be found in Table 10. The results of CLSOBBOA in solving FS problem. The results of CLSOB-BOA compared with original BOA, PSO, and GWO are shown in Tables 11–13 in terms of average fitness, feature size length, and classification accuracy. From these results, we can conclude the significant of CLSOBBOA in solving FS

**Table 8. Optimization results for pressure vessel design problem.**

| Algorithm | Optimization results | | | Cost | |
|---|---|---|---|---|---|
| | $T_s$ | $T_h$ | R | L | Cost |
| CLSOBBOA | 0.778 168 | 0.384 649 | 40.319 618 | 200 | 5885.332 773 |
| LSA | 0.843 656 | 0.417 020 | 43.712 767 | 40.363 464 | 6006.957 652 |
| SOS | 0.779 253 | 3.850 801 | 157.609 | 199.458 | 5889.984 071 |
| ABC | 7.781 687 | 3.846 492 | 40.319 620 | 200 | 5885.333 300 |
| GWO | 0.778 915 | 0.384 960 | 40.342 623 | 200 | 5889.412 437 |

**Table 9. Optimization results for speed reducer design problem.**

| Algorithm | Optimization results | | | | | | Cost | |
|---|---|---|---|---|---|---|---|---|
| | b | m | p | l1 | l2 | d1 | d2 | Cost |
| CLSOBBOA | 3.501260 | 0.7 | 17 | 7.380 | 7.83 | 3.33241 | 5.26345 | 2995.775 |
| GWO | 3.501591 | 0.7 | 17 | 7.391 | 7.82 | 3.35127 | 5.28074 | 2998.5507 |
| AMO | 3.506700 | 0.7 | 17 | 7.380 | 7.82 | 3.35784 | 5.27676 | 3001.944 |
| WCA | 3.500219 | 0.7 | 17 | 8.379 | 7.84 | 3.35241 | 5.28671 | 3005.222 |
| SOS | 3.538402 | 0.7 | 17 | 7.392 | 7.81 | 3.3580 | 5.28677 | 3002.928 |

**Table 10. Descriptions of datasets.**

| Symbol | Dataset | No. of features | No. of instances |
|--------|---------|-----------------|------------------|
| DS1 | Breastcancer | 10 | 699 |
| DS2 | BreastEW | 31 | 569 |
| DS3 | WineEW | 14 | 178 |
| DS4 | segment | 20 | 2310 |
| DS5 | Zoo | 17 | 101 |

**Table 11. Statistical mean fitness measure calculated for the compared algorithms on the different datasets.**

| Dataset | CLSOBBOA | BOA | PSO | GWO |
|---------|----------|-----|-----|-----|
| DS1 | 0.300 | 0.451 | 0.356 | 0.416 |
| DS2 | 0.025 | 0.056 | 0.042 | 0.056 |
| DS3 | 0.010 | 0.030 | 0.014 | 0.022 |
| DS4 | 0.025 | 0.043 | 0.033 | 0.045 |
| DS5 | 0.008 | 0.026 | 0.013 | 0.031 |

**Table 12. Average classification accuracy for the compared algorithms on the different datasets.**

| Dataset | CLSOBBOA | BOA | PSO | GWO |
|---------|----------|-----|-----|-----|
| DS1 | 0.987 | 0.940 | 0.988 | 0.978 |
| DS2 | 0.951 | 0.915 | 0.985 | 0.962 |
| DS3 | 0.999 | 0.981 | 0.996 | 0.992 |
| DS4 | 0.985 | 0.946 | 0.984 | 0.977 |
| DS5 | 0.999 | 0.981 | 0.996 | 0.996 |

**Table 13. Average selection size for the compared algorithms on the different datasets.**

| Dataset | CLSOBBOA | BOA | PSO | GWO |
|---------|----------|-----|-----|-----|
| DS1 | 3.4 | 3.8 | 3.6 | 4.6 |
| DS2 | 5.4 | 12.4 | 12.9 | 15.7 |
| DS3 | 2.6 | 5.2 | 3.7 | 6.1 |
| DS4 | 4.1 | 7.6 | 6.4 | 9.1 |
| DS5 | 3.1 | 6.1 | 4.3 | 6.5 |

# 4 Conclusion & future work

In this paper, a 3 variants of BOA algorithm have been introduced to improve its performance and preventing it from getting trapped in optimal subregion. These version merge the original BOA with Chaotic local search strategy and Opposition-based Learning concepts. The results show that the algorithm named CLSOBBOA have ranked first in more than half of CEC2014 benchmark functions. Although, the proposed algorithm tested using 4 different constrained engineering problems.

# 5 Algorithms codes

Codes used in this paper can be found from the following Links:

# 6 Appendix B

## 6.1 Welded beam design problem

Minimize: $f_1(x) = 1.10471 * x(1)^2 * x(2) + 0.04811 * x(3) * x(4) * (14.0 + * x(2))$

Subject to: $g_1(x) = \tau - 13600$

$g_2(x) = \sigma - 30000$

$g_3(x) = x(1) - x(4)$

$g_4(x) = 6000 - p$

Variable Range

$0.125 \leq x_1 \leq 5$

$0.1 \leq x_2 \leq 10$

$0.1 \leq x_3 \leq 10$

$0.125 \leq x_4 \leq 5$

## 6.2 Tension/Compression spring design problem

Minimize: $f(x) = (x_3 + 2)x_2 x_1^2$

Subject to: $g_1(x) = 1 - (x_2^3 x_3 / 71\,,785 x_1^4) \leq 0$

$g_2(x) = (4x_2^2 - x_1 x_2 / 12,566(x_2 x_1^3 - x_1^4) + (1/5108 x_1^2)) - 10 \leq 0$

$g_3(x) = 1 - (140.45 x_1 / x_2^2 x_3) \leq 0$

$g_4(x) = (x_2 + x_1)/1.5 - 1 \leq 0,$

Variable Range

$0.05 \leq x_1 \leq 2.00$

$0.25 \leq x_2 \leq 1.30$

$2.00 \leq x_3 \leq 15.00$

## 6.3 Pressure vessel design problem

Minimize: $f(x) = 0.6224 x_1 x_3 x_4 + 1.7781 x_2 x_3^2 + 3.1661 x_1^2 x_4 + 19.84 x_1^2 x_3$

Subject to: $g_1(x) = -x_1 + 0.0193x$

$g_2(x) = -x_2 + 0/00954 x_3 \leq 0$

$g_3(x) = -\pi x_3^2 x_4 - (4/3)\pi x_3^3 + 1,296,000 \leq 0$

$g_4(x) = x_4 - 240 \leq 0$

Variable Range

$0 \leq x_i \leq 100, \quad i = 1, 2$

$0 \leq x_i \leq 200, \quad i = 3, 4$

## 6.4 Speed reducer design problem

Minimize: $f(x) = 0.7854x_1x_2^2(14.9334x_3 + 3.3333333x_3^2 - 43.0934) + 0.7854(x_4x_6^2 + x_5x_7^2 - 1.508(x_6^2 + x_7^2)$

Subject to:

$$g_1 = \frac{27}{x_1x_2^2x_3} - 1 \leq 0$$

$$g_2 = \frac{397.5}{x_1x_2^2x_3} - 1 \leq 0$$

$$g_3 = \frac{1.93x_4^3}{x_2x_3x_7^4} - 1 \leq 0$$

$$g_4 = \frac{1.93x_5^3}{x_2x_3x_6^4} - 1 \leq 0$$

$$g_5 = \frac{1}{110x_6^3}\sqrt{\left(\left(\frac{745x_4}{x_2x_3}\right)^2 + 16.9X10^6\right)} - 1 \leq 0$$

$$g_6 = \frac{1}{85x_7^3}\sqrt{\left(\left(\frac{745x_4}{x_2x_3}\right)^2 + 157.5X10^6\right)} - 1 \leq 0$$

$$g_7 = \frac{x_2x_3}{40} - 1$$

$$g_8 = \frac{5x_2}{x_1} - 1$$

$$g_9 = \frac{x_1}{12x_2} - 1$$

Variable Range

$2.6 \leq x_1 \leq 3.6$

$0.7 \leq x_2 \leq 0.8$

$17 \leq x_3 \leq 28$

$7.3 \leq x_4 \leq 8.3$

$7.8 \leq x_5 \leq 8.3$

$2.9 \leq x_6 \leq 3.9$

$5 \leq x_7 \leq 5.5$

**6.4.1 Gear train design problem.** Minimize: $f(x) = \left(\frac{1}{6.931} - \frac{x_2x_3}{x_1x_4}\right)^2$

Variable Range

$12 \leq x_i \leq 60, \quad i = 1, 2, 3, 4$

## Supporting information

**S1 Data.**

(RAR)

**S2 Data.**
(RAR)

**S1 File.**
(DOCX)

## Author Contributions

**Conceptualization:** Adel Saad Assiri.

**Data curation:** Adel Saad Assiri.

**Formal analysis:** Adel Saad Assiri.

**Funding acquisition:** Adel Saad Assiri.

**Investigation:** Adel Saad Assiri.

**Methodology:** Adel Saad Assiri.

**Project administration:** Adel Saad Assiri.

**Resources:** Adel Saad Assiri.

**Software:** Adel Saad Assiri.

**Supervision:** Adel Saad Assiri.

**Validation:** Adel Saad Assiri.

**Visualization:** Adel Saad Assiri.

**Writing – original draft:** Adel Saad Assiri.

**Writing – review & editing:** Adel Saad Assiri.

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
