## [Decision Letter · Decision Letter 0]

24 Aug 2020

PONE-D-20-20957

On the performance improvement of Butterfly Optimization approaches for global optimization

PLOS ONE

Dear Dr. Assiri,

Thank you for submitting your manuscript to PLOS ONE. After careful consideration, we feel that it has merit but does not fully meet PLOS ONE’s publication criteria as it currently stands. Therefore, we invite you to submit a revised version of the manuscript that addresses the points raised during the review process.

We look forward to receiving your revised manuscript.

Kind regards,

Seyedali Mirjalili

Academic Editor

PLOS ONE

Journal Requirements:

3. We note that Figures in your submission contain copyrighted images.

All PLOS content is published under the Creative Commons Attribution License (CC BY 4.0), which means that the manuscript, images, and Supporting Information files will be freely available online, and any third party is permitted to access, download, copy, distribute, and use these materials in any way, even commercially, with proper attribution. For more information, see our copyright guidelines: http://journals.plos.org/plosone/s/licenses-and-copyright.

a.         You may seek permission from the original copyright holder of Figure(s) [#] to publish the content specifically under the CC BY 4.0 license.

4. We suggest that you include a Conclusions section at the end of your manuscript.

Reviewers' comments:

Reviewer's Responses to Questions

**Comments to the Author**

1. Is the manuscript technically sound, and do the data support the conclusions?

Reviewer #1: Yes

Reviewer #2: Yes

2. Has the statistical analysis been performed appropriately and rigorously? 

Reviewer #1: Yes

Reviewer #2: No

3. Have the authors made all data underlying the findings in their manuscript fully available?

Reviewer #1: Yes

Reviewer #2: Yes

4. Is the manuscript presented in an intelligible fashion and written in standard English?

Reviewer #1: Yes

Reviewer #2: Yes

5. Review Comments to the Author

Reviewer #1: This is a good work, but a number of major and minor amendments are required as follows:

* Potential applications of the proposed CLSOBBOA should be discussed

* There is no justification of the CLSOBBOA method. Why for this problem area, please discuss. There are many other similar methods in the literature in this area, so such a justification is required.

* There is no statistical test to judge about the significance of the CLSOBBOA’s results. Without such a statistical test, the conclusion cannot be supported.

* There is no discussion on the cost effectiveness of the proposed CLSOBBOA method. What is the computational complexity? What is the runtime? Please include such discussions. You can also use the big oh notation to show the computation complexity.

* To have an unbiased view in the paper, there should be some discussions on the limitations of the proposed CLSOBBOA method

* Analysis of the results is missing in the paper. There is a big gap between the results and conclusion. There should be the result analysis between these two sections. After comparing the methods, you have to be able to analyse the results and relate them to the structure of all algorithms. It would be interesting to have your thoughts on why the method works that way? Such analyses would be the core of your work where you prove your understanding of the reason behind the results. You can also link the findings to the hypotheses of the paper. Long story short, this paper requires a very deep analysis from different perspectives

* How do you ensure that the comparison between CLSOBBOA and the comparative methods is fair?

* The proposed CLSOBBOA method might be sensitive to the values of its main controlling parameter. How did you tune the parameters?

* The main solution is an optimization, but the literature review of other metaheuristics is missing. Please provide an in-depth review to show readers a big picture of this field with recent and popular algorithms.

Some cosmetic comments:

* Avoid using first person.

* Highlights are missing.

* Avoid using abbreviations and acronyms in title, abstract, headings and highlights.

* Please avoid having heading after heading with nothing in between, either merge your headings or provide a small paragraph in between.

* Abstract is too short. Abstract should have one sentence per each: context and background, motivation, hypothesis, methods, results, conclusions.

* The first time you use an acronym in the text, please write the full name and the acronym in parenthesis. Do not use acronyms in the title, abstract, chapter headings and highlights.

* The results should be further elaborated to show how they could be used for the real applications.

* The originality of the paper needs to be further clarified.

Reviewer #2: The author have introduced three improved versions of BOA to prevent the original algorithm from getting trapped in local optima and have a good balance between exploration and exploitation abilities. However, there are few points have been raised though out the review that need to be addressed by the author as below.

1. Potential applications of the proposed method should be discussed

2. There is no justification of the method. Why for this problem area, please discuss. There are many other similar methods in the literature in this area, so such a justification is required.

3. There is no statistical test to judge about the significance of the method’s results. Without such a statistical test, the conclusion cannot be supported.

4. There is no discussion on the cost effectiveness of the proposed method. What is the computational complexity? What is the runtime? Please include such discussions. You can also use the big oh notation to show the computation complexity.

5. To have an unbiased view in the paper, there should be some discussions on the limitations of the proposed method

6. The proposed method might be sensitive to the values of its main controlling parameter. How did you tune the parameters?

7. The given discussion on the obtained results should be further improved, for example, there were some noticeable performance improvements, but the reasons behind are not cohesively discussed, improvements were presented as a rule of thumb. The authors should make linkage between the achieved improvement and the proposed methodology to make kind of justifications on why the results are significant?

6. PLOS authors have the option to publish the peer review history of their article (what does this mean?). If published, this will include your full peer review and any attached files.

Reviewer #1: No

Reviewer #2: No

---

## [Author Response · Author response to Decision Letter 0]

7 Oct 2020

Reviewer #1

Reviewer Comments Authors Responses

This is a good work, but a number of major and minor amendments are required as follows:

1. Potential applications of the proposed CLSOBBOA should be discussed

2. There is no justification of the CLSOBBOA method. Why for this problem area, please discuss. There are many other similar methods in the literature in this area, so such a justification is required. 

3. There is no statistical test to judge about the significance of the CLSOBBOA’s results. Without such a statistical test, the conclusion cannot be supported.

4. There is no discussion on the cost effectiveness of the proposed CLSOBBOA method. What is the computational complexity? What is the runtime? Please include such discussions. You can also use the big oh notation to show the computation complexity.

5. To have an unbiased view in the paper, there should be some discussions on the limitations of the proposed CLSOBBOA method

6. Analysis of the results is missing in the paper. There is a big gap between the results and conclusion. There should be the result analysis between these two sections. 

7. How do you ensure that the comparison between CLSOBBOA and the comparative methods is fair?

8. The proposed CLSOBBOA method might be sensitive to the values of its main controlling parameter. How did you tune the parameters?

9. Avoid using first person. 

10. Highlights are missing. 

11. Avoid using abbreviations and acronyms in title, abstract, headings and highlights.

12. Please avoid having heading after heading with nothing in between, either merge your headings or provide a small paragraph in between. All the comments and suggestions have been considered, answered and new paragraphs have been added to the paper as follows.

1. Reply: CLSOBBOA has been already applied for FS as in Page 26-28.

2. Reply: illustration statement has been added into the introduction to shows the problem of the original BOA as shown in page 2. 

3. Reply: Wilicoxon Rank Sum have been used as shown in page 9.

4. Reply: CLSOBBOA complexity has been calculated as shown in page 3 and 6.

5. Reply: As stated in the manuscript. There is no algorithms can solve all optimization problems according to NFL as shown in page 2.

6. Reply: Results section has been modified and enhanced as in page 6-8.

7. Reply: as stated in the manuscript all algorithms has been executed on the same machine and repeated for 20 times

8. Reply: The Original paper argued that BOA parameter is selected after many experiments

9. Reply: All manuscript has been modified

10. Reply: Highlights has been added in separate file

11. Reply: All manuscript has been modified

12. Reply: some sentences have bben added between titles

---

## [Decision Letter · Decision Letter 1]

6 Nov 2020

On the performance improvement of Butterfly Optimization approaches for global optimization and Feature Selection

PONE-D-20-20957R1

Dear Dr. Assiri,

We’re pleased to inform you that your manuscript has been judged scientifically suitable for publication and will be formally accepted for publication once it meets all outstanding technical requirements.

Kind regards,

Seyedali Mirjalili

Academic Editor

PLOS ONE

Additional Editor Comments (optional):

Reviewers' comments:

Reviewer's Responses to Questions

**Comments to the Author**

1. If the authors have adequately addressed your comments raised in a previous round of review and you feel that this manuscript is now acceptable for publication, you may indicate that here to bypass the “Comments to the Author” section, enter your conflict of interest statement in the “Confidential to Editor” section, and submit your "Accept" recommendation.

Reviewer #1: (No Response)

2. Is the manuscript technically sound, and do the data support the conclusions?

Reviewer #1: (No Response)

3. Has the statistical analysis been performed appropriately and rigorously? 

Reviewer #1: (No Response)

4. Have the authors made all data underlying the findings in their manuscript fully available?

Reviewer #1: (No Response)

5. Is the manuscript presented in an intelligible fashion and written in standard English?

Reviewer #1: (No Response)

6. Review Comments to the Author

Reviewer #1: My comments have been addressed. Before uploading the final version or in the proof stage, please make sure to proofread the paper again to polish the language.

7. PLOS authors have the option to publish the peer review history of their article (what does this mean?). If published, this will include your full peer review and any attached files.

Reviewer #1: No

---

## [Editor Report · Acceptance letter]

19 Nov 2020

PONE-D-20-20957R1 

On the performance improvement of Butterfly Optimization approaches for global optimization and Feature Selection 

Dear Dr. Assiri:

I'm pleased to inform you that your manuscript has been deemed suitable for publication in PLOS ONE. Congratulations! Your manuscript is now with our production department. 

Kind regards, 

on behalf of

Prof. Seyedali Mirjalili 

Academic Editor

PLOS ONE